# Surge Phenomenon in Optimal Learning Rate and Batch Size Scaling

**Shuaipeng Li**[*,1,†], **Penghao Zhao**[*,1,2], **Hailin Zhang**[*,1,2], **Xingwu Sun**[*,1,3], **Hao Wu**[1],
**Dian Jiao**[1], **Weiyan Wang**[1], **Chengjun Liu**[1], **Zheng Fang**[1], **Jinbao Xue**[1], **Yangyu Tao**[1],
**Bin Cui**[2,4†], **Di Wang**[1,†]

[1] Tencent Hunyuan
[2] School of Computer Science & Key Lab of High Confidence
Software Technologies (MOE), Peking University
[3] University of Macau
[4] Institute of Computational Social Science, Peking University (Qingdao)

## Abstract

In current deep learning tasks, Adam-style optimizers—such as Adam, Adagrad, RMSprop, Adafactor, and Lion—have been widely used as alternatives to SGD-style optimizers. These optimizers typically update model parameters using the sign of gradients, resulting in more stable convergence curves. The learning rate and the batch size are the most critical hyperparameters for optimizers, which require careful tuning to enable effective convergence. Previous research has shown that the optimal learning rate increases linearly (or follows similar rules) with batch size for SGD-style optimizers. However, this conclusion is not applicable to Adam-style optimizers. In this paper, we elucidate the connection between optimal learning rates and batch sizes for Adam-style optimizers through both theoretical analysis and extensive experiments. First, we raise the scaling law between batch sizes and optimal learning rates in the "sign of gradient" case, in which we prove that the optimal learning rate first rises and then falls as the batch size increases. Moreover, the peak value of the surge will gradually move toward the larger batch size as training progresses. Second, we conduct experiments on various CV and NLP tasks and verify the correctness of the scaling law.

## 1 Introduction

Deep learning techniques, initiated by Stochastic Gradient Descent (SGD) learning on large datasets, have significantly revolutionized various real-world applications [1]. Over the past decade, numerous optimizers, such as momentum [2], Adagrad [3], ADADELTA [4], RMSprop [5], Adam [6], Adafactor [7], and Lion [8], have been introduced to stabilize the iterative learning process and expedite convergence. Among them, the Adam optimizer is the most widely used across various domains including Computer Vision (CV) [9–11], Natural Language Processing (NLP) [12–15] and many others [16, 17]. It retains the first and second moment information of parameters to facilitate adaptive learning step size. Unlike SGD-style optimizers that use the raw gradient to determine the learning direction and step size, Adam and its variants (Adagrad, RMSprop, Lion, etc.) employ the sign of gradient for this purpose, thereby ensuring greater robustness [18].

Beyond specific hyper-parameters in optimizer configurations, the batch size and the learning rate are the most critical hyperparameters influencing convergence. As the scale of training datasets in various

---

[*]Equal contribution. [†]Corresponding author.

38th Conference on Neural Information Processing Systems (NeurIPS 2024).

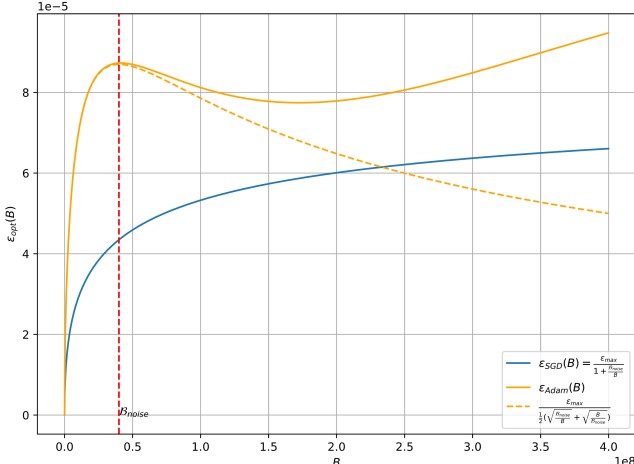

Figure 1: The relationship between the optimal learning rate and the batch size is different between Adam and SGD. The orange line represents the tendency of the optimal learning rate to converge to a non-zero value when the batch size is large enough.

workloads (e.g. CV [19, 20], NLP [21, 14], and others) continues to grow, there is an increasing demand for large batch size training across multiple data parallel workers. However, large batch training presents great challenges for robust training and meticulous tuning. The learning rate, which determines the actual step size in each learning iteration, is highly dependent on the batch size used. Prior research has explored methods to determine an optimal learning rate according to the batch size in scenarios utilizing SGD optimizers, including square root scaling [22], linear scaling [19, 23], and others [24]. Among these, the empirical model focusing on large batch training [24] yields convincing results in both theoretical and empirical contexts, proposing the following rule to depict the relationship between the optimal learning rate and the batch size:

$$\epsilon_{opt}(B) = \frac{\epsilon_{max}}{1 + \frac{\mathcal{B}_{noise}}{B}} \tag{1}$$

For Adam-style optimizers, though existing works also provide some approximation [24, 23], they fail to capture the true scaling law of optimal learning rates with batch sizes. For illustrative purposes, Figure 1 presents curves that simulate the optimal learning rates for the Adam optimizer. We find that, in scenarios involving small batch sizes, the optimal learning rate initially increases and then decreases, resembling a surge in the sea, as depicted by the dashed orange line. For large batch sizes, we identify a value to which the optimal learning rate converges. The solid orange line represents a schematic based on our findings for both small and large batch sizes, showing that the learning rate tends to rise initially, then decrease, and subsequently gradually ascend to asymptotically approach a stable value. For clarity in visualization, we have omitted the final asymptotic portion of the curve.

In this paper, we aim to elucidate and formalize the connection between optimal learning rates and batch sizes for Adam-style optimizers. By following the notation from the empirical model [24] and conducting a more in-depth theoretical analysis, we discover that the relationship between the optimal learning rate and the batch size in the above parameter update formular satisfies:

$$\epsilon_{opt}(B) = \frac{\epsilon_{max}}{\frac{1}{2}\left(\sqrt{\frac{\mathcal{B}_{noise}}{B}} + \sqrt{\frac{B}{\mathcal{B}_{noise}}}\right)}, \tag{2}$$

which differs from SGD, especially when the batch size is not too large. Here the meaning of $\mathcal{B}_{noise}$ is consistent with papers of scaling laws [24, 25], representing the trade-off point between training speed and data efficiency. When the batch size is equal to $\mathcal{B}_{noise}$, the optimal learning rate reaches a local maximum in accordance with Eq 2. Furthermore, we provide additional proof that when the batch size becomes significantly large, the optimal learning rate gradually converges to a non-zero value. We also prove that the previous conclusions about training speed and data efficiency are still valid for Adam-style optimizers, and the variable $\mathcal{B}_{noise}$ gradually increases as the training progresses. It is worth noting that when $B \ll \mathcal{B}_{noise}$, for SGD, the scaling law of optimal learning rates with

batch sizes transitions into linear scaling, consistent with previous conclusions [19, 23]:

$$\epsilon_{opt}(B) \approx \frac{\epsilon_{max}}{\mathcal{B}_{noise}} B; \tag{3}$$

while for Adam, the relationship transitions into square root scaling, aligning with previous approximations [23, 24]:

$$\epsilon_{opt}(B) \approx \frac{2\epsilon_{max}}{\sqrt{\mathcal{B}_{noise}}} \sqrt{B}. \tag{4}$$

In addition to theoretical analysis, our extensive empirical study on various CV and NLP workloads further validate the conclusions. The true optimal learning rate, across different Adam configurations, exhibits a clear downward trend after reaching its peak value as the batch size increases. This behavior contradicts previous research, but demonstrates the correctness and generalizability of our theory. The experiments also reveal a gradual increase in the variable $\mathcal{B}_{noise}$, corresponding to the peak optimal learning rate, as training progresses.

## 2 Theorems

### 2.1 Batch Size and Optimal Learning Rate

In this section, we theoretically derive the optimal learning rate for a given batch size. Initially, we introduce an approximation of Adam-style optimizers. In alignment with the insights elucidated in [26], a thorough examination of the Adam optimizer and its variants reveals that their primary distinction from SGD resides in the utilization of the gradient's sign for updates during each iteration, as opposed to the gradient itself:

$$\theta_{i+1} = \theta_i - \epsilon \cdot sign(G_{est}), \tag{5}$$

where $\theta_t$ is the parameter at time $t$, $G_{est}$ is the gradient estimated via mini-batch, and $\epsilon$ is the learning rate. As the batch size increases, the expected value of the update amount tends to saturate. For example, assuming that the mean value of the gradient is positive, when the accumulated gradient of the mini-batch is positive, increasing the batch size will have no contribution to the signed update amount. This is significantly different from the behavior of SGD where the larger the batch size, the more accurate the gradient estimate. In Appendix A, we provide a detailed discussion on this approximation for the Adam optimizer. Next, we derive the optimal learning rate that maximizes the loss improvement. And then we establish a lemma that addresses the optimal learning rate given an estimated mini-batch gradient:

**Lemma 1.** *Suppose that we are updating the parameter $\theta$ using the mini-batch gradient $V$, with the true gradient being $G$ and the true Hessian being $H$. Then the optimal learning rate that maximizes the decrease in loss is:*

$$\epsilon_{opt} \equiv argmax_\epsilon \, \mathbb{E}[\Delta L] = \frac{G^T \mathbb{E}[V]}{tr[H \cdot cov(V)] + \mathbb{E}[V]^T H \mathbb{E}[V]}, \tag{6}$$

*and the corresponding loss improvement $\Delta L$ is:*

$$\Delta L_{opt} = \frac{G^T \mathbb{E}[V]}{2} \epsilon_{opt}. \tag{7}$$

The proof is in Appendix B. Although our conclusion is based on an approximation, we adopt the equal sign here to simplify the analysis, following the notation used in previous work [24].

Now let's consider the case where $V = sign(G_{est})$, and assuming that the estimated gradient $G_{est}$ follows a Gaussian distribution. The Gaussian distribution assumption is motivated by the following: if the mini batch size is sufficiently large, we can invoke the Central Limit Theorem (CLT) and approximate the distribution as Gaussian - a common assumption in previous research [27–30]. Furthermore, our experimental results confirm that the gradient distributions closely approximate Gaussian distributions, as illustrated in Figure 8 of Appendix H. We have the following theorem:

**Theorem 2.** *Suppose the gradient of parameter $i$ for each sample follows a Gaussian distribution with mean $\mu_i$ and variance $\sigma_i^2$, the expected loss improvement is:*

$$\Delta L_{opt} = \frac{1}{2} \frac{\sum_i \sum_j \mathcal{E}_i \mathcal{E}_j \mu_i \mu_j}{\sum_i (1 - \mathcal{E}_i^2) H_{i,i} + \sum_i \sum_j \mathcal{E}_i \mathcal{E}_j H_{i,j}}, \tag{8}$$

*and the corresponding optimal learning rate is*

$$\epsilon_{opt} = \frac{\sum_i \mathcal{E}_i \mu_i}{\sum_i (1 - \mathcal{E}_i^2) H_{i,i} + \sum_i \sum_j \mathcal{E}_i \mathcal{E}_j H_{i,j}}, \tag{9}$$

*where $\mathcal{E}_i$ is a function (derived from the Gauss error function) with respect to the batch size $B$:*

$$\mathcal{E}_i(B) = \frac{2}{\sqrt{\pi}} \int_0^{\sqrt{\frac{B}{2}} \frac{\mu_i}{\sigma_i}} e^{-t^2} dt \approx \frac{\frac{\mu_i}{\sigma_i}}{\sqrt{\frac{\pi}{2B} + \left(\frac{\mu_i}{\sigma_i}\right)^2}}. \tag{10}$$

We prove the above theorem in Appendix C.

An important observation from the proof is that, not only is the covariance matrix of $sign(G_{est})$ related to $B$, but its expected value also depends on $B$. This implies that in Eq 6 the numerator is the first-order form of the function about $B$, and the denominator is the second-order form of the function about $B$:

$$\epsilon(B) = \frac{\beta f(B)}{f(B)^2 + \gamma} = \frac{\beta}{f(B) + \frac{\gamma}{f(B)}}. \tag{11}$$

Therefore, the conclusion in the case of Adam cannot be derived by simply following the form mentioned in [24]:

$$\epsilon(B) \neq \frac{\epsilon_*}{\left(1 + \frac{\mathcal{B}_{noise}}{B}\right)^\alpha}. \tag{12}$$

Then we aim to derive the specific expression for the optimal learning rate with respect to the batch size through the following theorems.

**Theorem 3.** *When $B \ll \frac{\pi \sigma_i^2}{2\mu_i^2}$, the optimal learning rate is a function with respect to batch size $B$:*

$$\epsilon_{opt}(B) \approx \frac{1}{\frac{1}{2}\left(\sqrt{\frac{\mathcal{B}_{noise}}{B}} + \sqrt{\frac{B}{\mathcal{B}_{noise}}}\right)} \frac{\sqrt{\frac{\mathcal{B}_{noise}}{2\pi}} \sum_i \frac{\mu_i^2}{\sigma_i}}{\sum_i H_{i,i}} \leq \frac{\sqrt{\frac{\mathcal{B}_{noise}}{2\pi}} \sum_i \frac{\mu_i^2}{\sigma_i}}{\sum_i H_{i,i}}, \tag{13}$$

*where $\mathcal{B}_{noise}$ is a variable unrelated to batch size $B$:*

$$\mathcal{B}_{noise} = \frac{\pi \sum_i H_{i,i}}{2 \sum_i \sum_j \begin{cases} \frac{\mu_i \mu_j}{\sigma_i \sigma_j} & i \neq j \\ 0 & i = j \end{cases} H_{i,j}}. \tag{14}$$

*Defining $B_{peak}$ as the batch size at which the optimal learning rate reaches a peak value, it is obvious that:*

$$B_{peak} = \mathcal{B}_{noise}. \tag{15}$$

*The peak value is:*

$$\epsilon_{max} = \frac{\sqrt{\frac{\mathcal{B}_{noise}}{2\pi}} \sum_i \frac{\mu_i^2}{\sigma_i}}{\sum_i H_{i,i}}. \tag{16}$$

We prove the theorem in Appendix D. From the theorem we can finally get Eq 2, which implies that there is an interval where the batch size becomes larger and the optimal learning rate needs to be reduced. Considering that $\frac{\pi \sigma_i^2}{2\mu_i^2}$ is much larger than normal batch sizes in research and industry (as shown in Figure 2), this theorem can cover most of the scenarios. To make the conclusion more comprehensive, we also derive the following theorem:

**Theorem 4.** *When $B \gg \frac{\pi \sigma_i^2}{2\mu_i^2}$, the optimal learning rate becomes:*

$$\epsilon_{opt} = \frac{\sum_i |\mu_i|}{\sum_i \sum_j sign(\mu_i) sign(\mu_j) H_{i,j}}. \tag{17}$$

We prove the theorem in Appendix E.

Therefore, when B increases infinitely, the optimal learning rate will eventually converge to a non-zero value. If we make an (unrealistic) assumption that $\frac{\mu_i}{\sigma_i} \approx sign(\mu_i)$, we will find that the lower bound of $\epsilon_{max}$ in Theorem 3 will become the one in Theorem 4, which means that the local peak value of the optimal learning rate is larger than the final convergence value. However, considering that the variance of the gradient in the later stages of training is very small, which makes the above conclusion $\frac{\mu_i}{\sigma_i} \approx sign(\mu_i)$ difficult to establish, so the stable value in the later stages of training is more likely to exceed the local maximum. We provide a reference curve in Figure 1.

## 2.2 Data/Time Efficiency Trade-off

Following the empirical model for large-batch training [24], we also review the trade-off between data and time efficiency during batch size selection. We have the following theorem:

**Theorem 5.** *When $B \ll \frac{\pi \sigma_i^2}{2\mu_i^2}$, the derived loss improvement with respect to the batch size is*

$$\Delta L_{opt}(B) = \frac{\Delta L_{max}}{1 + \frac{\mathcal{B}_{noise}}{B}}, \tag{18}$$

*where $\Delta L_{max}$ is defined as*

$$\Delta L_{max} = \frac{\sum_i \sum_j \frac{\mu_i^2 \mu_j^2}{\sigma_i \sigma_j}}{2 \sum_i \sum_j \begin{cases} \frac{\mu_i \mu_j}{\sigma_i \sigma_j} & i \neq j \\ 0 & i = j \end{cases} H_{i,j}}, \tag{19}$$

We prove the theorem in Appendix F. This result aligns with the conclusion drawn in the SGD situation [24], indicating that many related conclusions also remain valid.

It has been concluded in previous work [24] that, when using the SGD optimizer with the same form as Eq 18, the relationship between training speed (number of steps $S$) and data efficiency (number of samples $E$) is given by:

$$\left(\frac{S}{S_{min}} - 1\right)\left(\frac{E}{E_{min}} - 1\right) = 1. \tag{20}$$

Here $S_{(min)}$ represents training speed, the actual (minimum) possible number of steps taken to reach a specified model performance; and $E_{(min)}$ represents data efficiency, the actual (minimum) possible number of training examples processed to reach that same level of performance. For more details, please refer to the Eq 2.11 and the Appendix D in [24]. Additionally, as referenced in the Eq 2.12 in [24] and Eq 1.4 in [25], $\mathcal{B}_{noise}$ is the balance point between training speed and data efficiency:

$$\mathcal{B}_{noise} \approx \mathcal{B}_{crit} = \frac{E_{min}}{S_{min}} \approx \frac{B_*}{L^{\frac{1}{\alpha_B}}}. \tag{21}$$

Since in Adam optimizer we arrive at the same Eq 18 as in SGD optimizer, the above equations 20 and 21 still hold. In Adam scenarios, $B_{peak} = \mathcal{B}_{noise}$ is not only the local maximum of the optimal learning rate, but also the balance point between training speed and data efficiency. Moreover, as training progresses and the loss decreases, according to Eq 21, $B_{peak}$ will gradually becomes larger.

## 2.3 Summary

In this section, we have drawn several conclusions from our theoretical analysis, which are summarized as follows:

1. As the batch size increases, the optimal learning rate demonstrates a decreasing trend within a specified range (Eq 2).

2. The batch size that corresponds to the local maximum optimal learning rate is consistent with the balance point of training speed and data efficiency (Eq 21). As the training progresses and the loss decreases, $B_{peak}$ will gradually becomes larger.

# 3 Experiments

In this section, we carry out a series of experiments to corroborate the theoretical scaling law we proposed in Section 2 and detail the experimental workloads and configurations in Section 3.1. The process for deriving the estimated variables from our theory is elucidated in Section 3.2. We also showcase and dissect the applicability of our scaling law across a variety of workloads in Section 3.3.

## 3.1 Experimental Setup

**Workloads.** In our empirical study, we incorporate 4 open-source workloads that are extensively utilized: (1) training a 5-layer CNN model on the Fashion-MNIST [31], which is a typical CV test case to start with. It consists of 60000 28x28 grayscale images in 10 classes; (2) training a ResNet-18 model [32] on the Tiny-ImageNet dataset [33], which contains 100000 images of 200 classes (500 for each class) downsized to 64×64 colored images. In each epoch we train the model with random 10k samples to reduce the overall complexity; (3) training a dense Transformer model [12] (simplified DistilGPT2 [34]) on the ELI5-Category dataset [35], which is a smaller but newer and categorized version of the original ELI5 dataset [36]. It contains 10.5k complex, diverse questions that require explanatory multi-sentence answers. (4) training a fine-grained Mixture-of-Experts (MoE) model, similar in structure to Mistral-MoE [37] but with shared experts [38], on the RedPajama-v2 dataset [39], which contains 30 trillion filtered and deduplicated tokens (100+ trillions raw) from 84 CommonCrawl dumps covering 5 languages, along with 40+ pre-computed data quality annotations. These workloads are popular in both academia and industry, covering typical deep learning tasks in the domains of CV and NLP.

**Batch sizes and learning rates.** To showcase the optimal learning rate for each batch size configuration, we leverage a grid-search-style experiments set. Each point in the grid search corresponds to a certain round with the same configuration but a different random number seed. The start point, stop point, and the interval of different workloads are listed in Table 1. In NLP tasks, the term "batch size" refers to the number of tokens in a batch, as practiced in related works [25].

Table 1: Grid search configurations.

| Workload | Adam | | Learning Rate | | | Batch Size | | | Round |
|----------|------|------|-------|------|------|-------|------|------|-------|
| | $\beta_1$ | $\beta_2$ | Start | Stop | Step | Start | Stop | Step | |
| CNN | 0.9 | 0.999 | 1e-4 | 1e-3 | 1e-4 | 1 | 12 | 1 | 100 |
| CNN | 0.9 | 0.999 | 1e-4 | 1e-3 | 1e-4 | 64 | 1164 | 100 | 100 |
| DistilGPT2 | 0.9 | 0.999 | 1e-5 | 1.09e-3 | 1.2e-4 | 4 | 114 | 10 | 30 |
| DistilGPT2 | 0.0 | 0.0 | 1e-5 | 5.5e-4 | 6e-5 | 4 | 114 | 10 | 30 |
| ResNet18 | 0.0 | 0.0 | 1e-4 | 7.876e-4 | 7.65e-5 | 16 | 376 | 33 | 100 |
| MoE | 0.9 | 0.999 | 2e-6 | 6e-5 | 2e-6 | 192k | 12M | 1.2M | 10 |

**Hyper-parameters.** Since we derive the theorems on Adam-style optimizers, we conduct experiments using the Adam optimizer. We experiment on both the "sign of gradient" configuration ($\beta_1 = 0$, $\beta_2 = 0$) and the default hyper-parameters ($\beta_1 = 0.9$, $\beta_2 = 0.999$), as shown in Table 1.

**Hardware environment.** We execute each round of experiments utilizing an NVIDIA A100 card. The training time of each round for the datasets are approximately 1 hour for Fashion-MNIST, 1.5 hours for TinyImageNet, 2 hours for ELI5-Category and 11 hours for RedPajama-v2. Given our primary focus on the convergence process, the specific hardware environment does not matter in our experiments. Our theoretical and empirical findings can be generalized to other hardware settings. Additionally, some system optimizations [40–43] are also beneficial to enhancing training efficiency.

## 3.2 Variable Estimation

We try to estimate the value of $\mathcal{B}_{noise}$ and the expectation of $\epsilon_{max}$ through curve fitting. After using Eq 21 to simplify Eq 20 (see Appendix G for details), we can record the actual possible number of steps taken $S$ and the actual possible number of training examples processed $E$ to reach a specified level of performance corresponding to the optimal learning rate of each batch size in the grid search results, and then perform linear fitting to obtain the estimated value of $\mathcal{B}_{noise}$:

$$\frac{1}{S} = -\mathcal{B}_{noise}\frac{1}{E} + \frac{1}{S_{min}} \tag{22}$$

Subsequently, we use the optimal learning rate and batch size of the grid search results to estimate the max optimal learning rate of Adam-style $\mathbb{E}[\epsilon_{max}]_{Adam}$:

$$\mathbb{E}[\epsilon_{max}]_{Adam} = \mathbb{E}\left[\frac{\epsilon_{opt}}{2}\left(\sqrt{\frac{\mathcal{B}_{noise}}{B}} + \sqrt{\frac{B}{\mathcal{B}_{noise}}}\right)\right] \tag{23}$$

and SGD-style $\mathbb{E}[\epsilon_{max}]_{SGD}$:

$$\mathbb{E}[\epsilon_{max}]_{SGD} = \mathbb{E}\left[\epsilon_{opt}\left(1 + \frac{\mathcal{B}_{noise}}{B}\right)^{\alpha}\right] \tag{24}$$

Previous research [24] represents the SGD optimizer and the Adam optimizer using Eq 24 with $\alpha = 1$ and $\alpha = 0.5$, respectively. We include these fitted curves as comparisons in the following section.

While we use grid search to estimate the value of $\mathcal{B}_{noise}$, in practice we can efficiently approximate it using the scaling law from previous studies [24, 25], where $\mathcal{B}_{noise} \approx B_{crit} = \frac{B_*}{L^{1/\alpha}}$. With this approximation, we only need a simple search for a pair of (batch size, optimal learning rate) to determine the final hyperparameter $\epsilon_{max}$. Therefore, the costly grid-search can be avoided.

### 3.3 Results

Following Section 3.2, we first estimate the variables and fit the curves using observations, then conduct grid-search-style experiments for learning rates and batch sizes.

Figure 2, 3, 4, 5 illustrate the experimental results of CNN-FashionMNIST, ResNet18-TinyImageNet, DistilGPT2-ELI5Category and MoE-RedPajama-v2, respectively. Each figure is divided into two parts: the left subfigure illustrates the grid-search results for batch sizes and learning rates, and these data points are utilized to fit the curve of Eq 22 in the right subfigure. In order to estimate the variables, we train models from scratch using different learning rates and batch sizes, then record the number of steps $S$ and examples $E$ in each experiment to achieve an equivalent training loss. Using the recorded $S$ and $E$, we fit the curve in the right subfigure and obtain the estimated $\mathcal{B}_{noise}$. In the left subfigure, upon achieving the desired training loss, all experiments continue to train the same number of steps. Any subsequent decrease in training loss is represented through different colors, as indicated in the color bar. For each batch size, we highlight the optimal learning rate that results in the most significant reduction in training loss. We also plot the batch size $\mathcal{B}_{noise}$ that corresponds to the peak optimal learning rate, the fitted SGD curves with $\alpha = 0.5$ and $\alpha = 1$ as derived from previous research [24], and the fitted Adam curve as derived from our theorems.

For the CNN-FashionMNIST workload, we train exactly 10 more step after achieving the desired training loss. As shown in Figure 2(a), the batch size bound $\frac{\pi\sigma_i^2}{2\mu_i^2}$ for Theorem 3 is around 800 in this task. Given the simplicity of the CNN-FashionMNIST workload, commonly-used batch sizes are usually smaller than the batch size bound. We plot the situations corresponding to Theorem 3 and Theorem 4 in Figure 2(b) and 2(c), respectively. In both cases, the trend predicted by our theory is consistent with the actual optimal learning rate performance, showing a declining range at small batch sizes and a saturation range at large batch sizes.

For the ResNet18-TinyImageNet workload, we train 50 more steps after achieving the desired training loss. We plot the figures for Theorem 3 at different achieved training losses, which represent the progress of training, as shown in Figure 3. The observed optimal learning rates primarily exhibit a downward trend after the batch size exceeds the estimated $\mathcal{B}_{noise}$. Although the SGD curve with $\alpha = 0.5$, which is claimed by [24] to represent the Adam optimizer, serves as a good approximation in certain cases, it fails to capture the peak optimal learning rate as our Adam curve does. Comparing the red dashed lines in different figures, we can see that the estimated $\mathcal{B}_{noise}$ gradually increases as the training progresses (i.e. training loss decreases), which corroborates the second conclusion in Section 2.3.

For the DistilGPT2-Eli5Category workload, we train 50 more steps after achieving the desired training loss. As shown in Figure 4, we test on two distinct Adam configurations for Theorem 3: the first with $\beta_1 = 0.0$, $\beta_2 = 0.0$, and the second with $\beta_1 = 0.9$, $\beta_2 = 0.999$. In both configurations, promising learning rates that lead to a substantial decrease in loss are consistent with our Adam curve. It is worth noting that another curve, SGD with $\alpha = 0.5$ [24], also provides a suitable approximation in this scenario. To more clearly demonstrate the accuracy of our theoretical predictions, we present

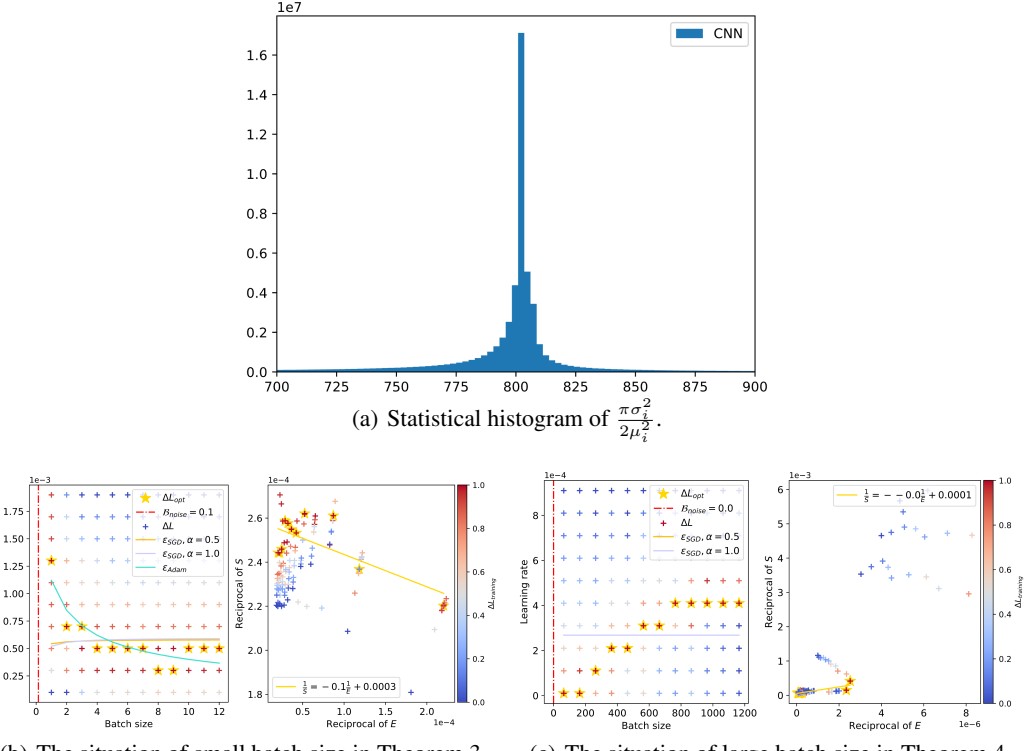

(a) Statistical histogram of $\frac{\pi\sigma_i^2}{2\mu_i^2}$.

(b) The situation of small batch size in Theorem 3. (c) The situation of large batch size in Theorem 4.

Figure 2: Batch size versus optimal learning rate within the context of CNN trained on FashionMNIST.

detailed results from a finer-grained grid search in Figure 6 of Appendix H. These experiments demonstrate that our theorems can be generalized to different optimizer configurations, validating the analysis in Appendix A.

For the sparse MoE model using the RedPajama-v2 dataset, we train 300 more steps after achieving the desired training loss. Figure 5 demonstrates that our predictions on optimal learning rate are both accurate and appropriate.

In addition to the above workloads, we also conduct an analysis of experimental results from third parties, confirming that our conclusions remain valid. Detailed results are presented in Figure 7 of Appendix H.

## 4 Discussion

We have carried out empirical studies on representative workloads using the Adam optimizer. Our investigation into the scaling laws of learning rates relative to batch sizes has provided deeper insights into the training dynamics of deep learning models. This understanding can help fine-tune hyperparameters, enhance convergence speeds, and circumvent exhaustive grid searches. By leveraging prior knowledge that the optimal learning rate decreases after reaching a peak, researchers and engineers can more effectively adjust the learning rate to achieve efficient training outcomes.

In real-world applications, there are numerous different learning workloads [44–47]. Other factors, beyond the scope of this paper, may influence the learning process - the specific optimizer used, weight decay, gradient clipping, etc. While we assert that our theorem can be applied to numerous practical scenarios, it may not fully encompass all situations involving intricate training configurations.

As one of our conclusions points out, the variable $\mathcal{B}_{noise}$ will gradually increases as the training progresses. It is natural to implement adaptive learning rates (and batch sizes) if possible, to speed up the training process. As mentioned in [24], using adaptive batch sizes and warmed-up learning rates brings considerable benefits. Fully exploring the potential of batch size and learning rate scheduling requires meticulous design, which we leave as future work.

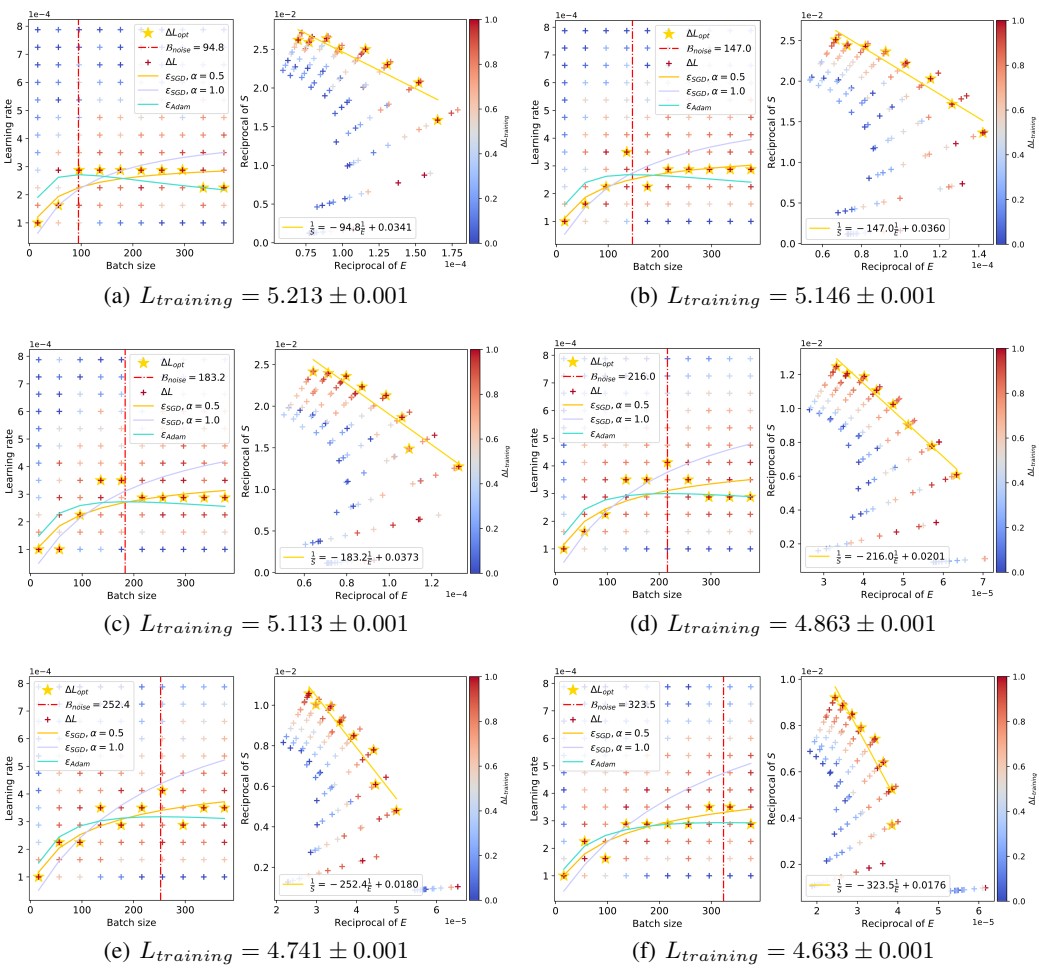

Figure 3: The relationship between batch sizes and optimal learning rates within the context of ResNet-18 trained on TinyImageNet. The red dashed line accurately predicts the peak value, and as the training loss decreases, the peak value gradually shifts to the right.

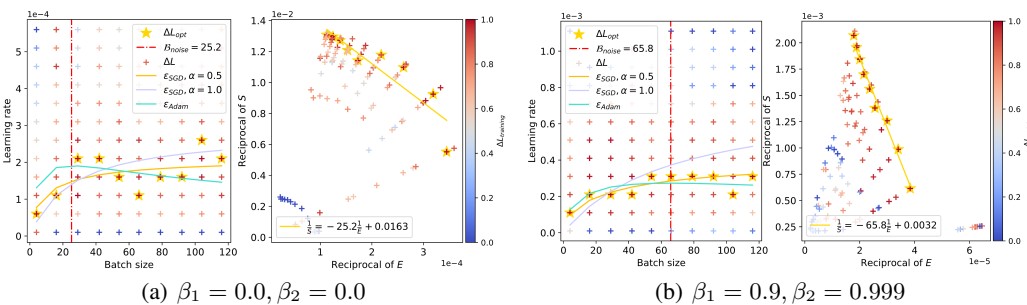

Figure 4: The relationship between batch sizes and optimal learning rates within the context of DistilGPT2 trained on Eli5Category.

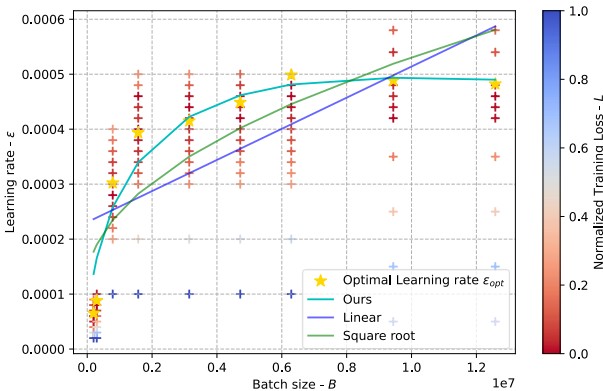

Figure 5: Grid search results for the MoE [37, 38] structure model.

Our theory is based on the quadratic approximation of the loss function. Experimental results demonstrate that conclusions drawn from this second-order expansion effectively predict the surge phenomenon observed in most mainstream scenarios. However, recent studies [48–50] have proposed that quadratic approximations do not accurately capture the loss in scenarios involving large learning rates. We acknowledge the potential benefits of exploring higher-order approximations and consider this a promising direction for future research.

# 5    Related Work

Aiming to accelerate convergence, our work analyzes the scaling law of optimal learning rates with respect to batch sizes for Adam-style optimizers. Numerous related studies have been proposed to enhance the convergence of deep learning tasks by investigating optimal learning rates, developing new optimizers, and analyzing gradient noise.

Earlier works have proposed various scaling laws to tune learning rates for SGD-style optimizers, including square root scaling [22], linear scaling [19, 23], and others [24]. They also obtained a scaling law for Adam-style optimizers [24, 23] through approximation, revealing a square root-like relationship where the optimal learning rate monotonically increases with the batch size. However, as illustrated in Section 1 and 2, their analysis holds only for small batch sizes, whereas the true scaling law exhibits greater complexity, with the optimal learning rate reaching a peak value at a balanced batch size.

There are many meticulously designed optimizers for various tasks and scenarios. First-order optimizers dominate nowadays deep learning models, including adaptive methods [3–7], sign-based methods [18, 8], layer-wise methods (for large-batch training) [51, 52]. Second-order optimizers [53–55], though with stronger theoretical guarantees, are not efficient for large-scale models due to quadratic complexity with respect to the number of parameters. Despite the emergence of new optimizers, empirical evidence confirms that Adam has remained the most widely used and effective optimizer over the past decade.

Our analysis is inspired by the empirical model of large-batch training [24], which predicts the useful batch size using the gradient noise scale. Gradient noise can help with learning rate determination [56], batch size selection [57, 58], and gaining deeper insights into the convergence process [59–62].

# 6    Conclusion

In this paper, we established a scaling law between optimal learning rates and batch sizes for Adam-style optimizers. We theoretically proved that the optimal learning rate initially increases and then decreases as the batch size grows, and that the peak value of the surge represents a trade-off point between training speed and data efficiency. Through extensive experiments, we validated our theory on diverse deep learning models and datasets.

## Acknowledgments and Disclosure of Funding

This work is supported by National Science and Technology Major Project (2022ZD0116315), National Natural Science Foundation of China (U22B2037, U23B2048), Beijing Municipal Science and Technology Project (Z231100010323002), research grant No. SH-2024JK29, PKU-Tencent joint research Lab, and High-performance Computing Platform of Peking University. Bin Cui, Shuaipeng Li and Di Wang are the corresponding authors.

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

## A Parameter Update Amount in the Adam Optimizer

The update amount in the Adam optimizer consists of two parts, the first moment:

$$m_t = \beta_1 m_{t-1} + (1 - \beta_1) G_{est,t}$$
$$\widehat{m_t} = \frac{m_t}{1 - \beta_1^t} \tag{25}$$

and the second moment:

$$v_t = \beta_2 v_{t-1} + (1 - \beta_2) G_{est,t}^2$$
$$\widehat{v_t} = \frac{v_t}{1 - \beta_2^t} \tag{26}$$

where $m_0 = 0$ and $v_0 = 0$. The final update amount is

$$V = \frac{\widehat{m_t}}{\sqrt{\widehat{v_t}} + \epsilon_{Adam}} = \frac{\frac{1-\beta_1}{1-\beta_1^t} \sum_i^t \beta_1^{t-i} G_{est,i}}{\sqrt{\frac{1-\beta_2}{1-\beta_2^t} \sum_i^t \beta_2^{t-i} G_{est,i}^2} + \epsilon_{Adam}} \tag{27}$$

After ignoring the role of $\epsilon_{Adam}$, when $\beta_1 \to 1$ and $\beta_2 \to 1$, Eq 27 transforms to

$$V = \frac{\frac{\sum_i^t G_{est,i}}{t}}{\sqrt{\frac{\sum_i^t G_{est,i}^2}{t}}} = \frac{\mathbb{E}_t[G_{est}]}{\sqrt{\mathbb{E}_t[G_{est}^2]}} = \frac{sign(\mathbb{E}_t[G_{est}])}{\sqrt{1 + \frac{var_t(G_{est})}{\mathbb{E}_t[G_{est}]^2}}} \tag{28}$$

The equation is obtained by using $var_t(G_{est}) = \mathbb{E}_t[G_{est}^2] - \mathbb{E}_t[G_{est}]^2$. Note that the expected value $\mathbb{E}_t[G_{est}]$ is over the iteration distribution, not over the data distribution. Obviously, when the variance of $G_{est}$ is small, the update amount is approximately $sign(G_{est})$.

On the other hand, when $\beta_1 \to 0$ and $\beta_2 \to 0$, Eq 27 can be simplified to

$$V = \frac{G_{est}}{\sqrt{G_{est}^2}} = sign(G_{est}) \tag{29}$$

Therefore, we can approximate the parameter update amount of the Adam optimizer as $sign(G_{est})$, without affecting the theoretical conclusion.

## B Proof of Lemma 1

*Proof.* We can perturb the parameters $\theta$ by some vector $V$ with learning rate $\epsilon$, and approximate the true loss using a quadratic expansion in terms of $\epsilon$ via Taylor expansion:

$$L(\theta - \epsilon \cdot V) \approx L(\theta) - \epsilon G^T V + \frac{1}{2} \epsilon^2 V^T H V. \tag{30}$$

Consider the expected value of loss improvement over a data distribution $\rho(x)$ over data points $x$:

$$\mathbb{E}[\Delta L] = \mathbb{E}[L(\theta) - L(\theta - \epsilon \cdot V)] \approx \epsilon G^T \mathbb{E}[V] - \frac{1}{2} \epsilon^2 \mathbb{E}[V^T H V]. \tag{31}$$

By maximizing the expected loss improvement, we obtain the optimal learning rate in Eq 6 and the optimal loss improvement in Eq 7. □

## C Proof of Theorem 2

*Proof.* Consider a random variable $x \sim \mathcal{N}(\mu, \sigma^2)$, let $y = \frac{x-\mu}{\sigma}$ and $z = -\frac{\mu}{\sigma}$, then

$$\begin{aligned}
\mathbb{E}[sign(x)] &= \int_{-\infty}^{\infty} sign(x) \frac{1}{\sigma\sqrt{2\pi}} e^{-\frac{(x-\mu)^2}{2\sigma^2}} dx \\
&= \frac{1}{\sqrt{2\pi}} \int_{-\infty}^{\infty} sign(\mu + \sigma y) e^{-\frac{y^2}{2}} dy \\
&= \frac{1}{\sqrt{2\pi}} \left( \int_z^{\infty} e^{-\frac{y^2}{2}} dy - \int_{-\infty}^z e^{-\frac{y^2}{2}} dy \right) \\
&= (1 - \Phi(z)) - \Phi(z) = erf\left( \frac{\mu}{\sqrt{2}\sigma} \right)
\end{aligned} \tag{32}$$

and the variance is

$$var(sign(x)) = \mathbb{E}[sign(x)^2] - \mathbb{E}[sign(x)]^2 = 1 - erf\left(\frac{\mu}{\sqrt{2}\sigma}\right)^2, \tag{33}$$

where $\Phi$ represents the cumulative distribution function of the standard normal distribution, and $erf$ is the Gauss error function that is defined as $erf(x) = \frac{2}{\sqrt{\pi}}\int_0^x e^{-t^2}dt$.

Given that the gradient $G_x$ of any data point $x$ in the data distribution $\rho(x)$ relative to a certain parameter $\theta_i$ follows a Gaussian distribution with mean $\mu_i$ and variance $\sigma_i^2$, then

$$G_{est}(\theta_i) = \frac{1}{B}\sum_{}^{B} G_x(\theta_i) \sim \mathcal{N}\left(\mu_i, \frac{\sigma_i^2}{B}\right). \tag{34}$$

Therefore, when $V = sign(G_{est})$, using Eq 32 and Eq 33 we can get the expectation

$$\mathbb{E}[V] = \begin{pmatrix} \vdots \\ erf\left(\sqrt{\frac{B}{2}}\frac{\mu_i}{\sigma_i}\right) \\ \vdots \end{pmatrix}, \tag{35}$$

and the covariance matrix

$$cov(V) = \begin{pmatrix} \ddots & & 0 \\ & 1 - erf\left(\sqrt{\frac{B}{2}}\frac{\mu_i}{\sigma_i}\right)^2 & \\ 0 & & \ddots \end{pmatrix}. \tag{36}$$

Given that the real gradient satisfies:

$$G = \begin{pmatrix} \vdots \\ \mu_i \\ \vdots \end{pmatrix}, \tag{37}$$

and define $\mathcal{E}_i$ as a function with respect to the token batch size $B$, based on the Gauss error function:

$$\mathcal{E}_i(B) = erf\left(\sqrt{\frac{B}{2}}\frac{\mu_i}{\sigma_i}\right) = \frac{2}{\sqrt{\pi}}\int_0^{\sqrt{\frac{B}{2}}\frac{\mu_i}{\sigma_i}} e^{-t^2}dt, \tag{38}$$

substituting the expectation and the variance of $V$ in Eq 6 and Eq 7 from Lemma 1 using the above equations, we can get Eq 8 and Eq 9 respectively.

To simplify the subsequent computation, we approximate the function $\mathcal{E}_i$ by other sigmoid-like analytical forms. Specifically, we find that $\mathcal{E}_i(B) \approx \frac{\frac{\mu_i}{\sigma_i}}{\sqrt{\frac{\pi}{2B} + \left(\frac{\mu_i}{\sigma_i}\right)^2}}$, which results in Eq 10.

$\square$

## D   Proof of Theorem 3

*Proof.* When $B \ll \frac{\pi\sigma_i^2}{2\mu_i^2}$, Eq 10 reduces to:

$$\mathcal{E}_i(B) \approx \sqrt{\frac{2B}{\pi}}\frac{\mu_i}{\sigma_i}. \tag{39}$$

Substituting the function $\mathcal{E}_i$ in Eq 9 using the above equation, and defining $\mathcal{B}_{noise}$ and $\epsilon_{max}$ as in Eq 14 and 16, we can deduce the relationship between the optimal learning rate and the token batch size:

$$\epsilon_{opt}(B) = \frac{\sum_i \sqrt{\frac{2B}{\pi}} \frac{\mu_i}{\sigma_i} \mu_i}{\sum_i H_{i,i} + \frac{2B}{\pi} \sum_i \sum_j \begin{cases} \frac{\mu_i \mu_j}{\sigma_i \sigma_j} & i \neq j \\ 0 & i = j \end{cases} H_{i,j}}$$

$$= \frac{\frac{\sum_i \sqrt{\frac{2B}{\pi}} \frac{\mu_i^2}{\sigma_i}}{\sum_i H_{i,i}}}{1 + B \frac{2 \sum_i \sum_j \begin{cases} \frac{\mu_i \mu_j}{\sigma_i \sigma_j} & i \neq j \\ 0 & i = j \end{cases} H_{i,j}}{\pi \sum_i H_{i,i}}}$$

$$= \frac{\sqrt{B}}{1 + \frac{B}{\mathcal{B}_{noise}}} \frac{\sqrt{\frac{2}{\pi}} \sum_i \frac{\mu_i^2}{\sigma_i}}{\sum_i H_{i,i}}$$ (40)

$$= \frac{\frac{\sqrt{\mathcal{B}_{noise}}}{2}}{\frac{1}{2}\left(\sqrt{\frac{\mathcal{B}_{noise}}{B}} + \sqrt{\frac{B}{\mathcal{B}_{noise}}}\right)} \frac{\sqrt{\frac{2}{\pi}} \sum_i \frac{\mu_i^2}{\sigma_i}}{\sum_i H_{i,i}}$$

$$= \frac{\epsilon_{max}}{\frac{1}{2}\left(\sqrt{\frac{\mathcal{B}_{noise}}{B}} + \sqrt{\frac{B}{\mathcal{B}_{noise}}}\right)} \leq \epsilon_{max}.$$

It should be noted that since $\mathcal{B}_{noise} > 0$ and $\epsilon_{max} > 0$, then both $\sum_i H_{i,i}$ and $\sum_i \sum_j \begin{cases} \frac{\mu_i \mu_j}{\sigma_i \sigma_j} & i \neq j \\ 0 & i = j \end{cases} H_{i,j}$ are greater than 0. Therefore, $\epsilon_{max}$ can be expressed as a form that does not contain $\mathcal{B}_{noise}$:

$$\epsilon_{max} = \frac{\sqrt{\frac{\mathcal{B}_{noise}}{2\pi}} \sum_i \frac{\mu_i^2}{\sigma_i}}{\sum_i H_{i,i}}$$

$$= \frac{\sqrt{\sum_i H_{i,i}}}{2\sqrt{\sum_i \sum_j \begin{cases} \frac{\mu_i \mu_j}{\sigma_i \sigma_j} & i \neq j \\ 0 & i = j \end{cases} H_{i,j}}} \frac{\sum_i \frac{\mu_i^2}{\sigma_i}}{\sum_i H_{i,i}}$$

$$= \frac{\sum_i \frac{\mu_i^2}{\sigma_i}}{2\sqrt{\sum_i \sum_j \begin{cases} \frac{\mu_i \mu_j}{\sigma_i \sigma_j} & i \neq j \\ 0 & i = j \end{cases} H_{i,j} \cdot \sqrt{\sum_i H_{i,i}}}}.$$ (41)

$$\geq \frac{\sum_i \frac{\mu_i^2}{\sigma_i}}{\sum_i \sum_j \begin{cases} \frac{\mu_i \mu_j}{\sigma_i \sigma_j} & i \neq j \\ 1 & i = j \end{cases} H_{i,j}}$$

The last inequality is derived using the AM–GM inequality ($a^2 + b^2 \geq 2ab$).

□

# E  Proof of Theorem 4

*Proof.* When $B \gg \frac{\pi \sigma_i^2}{2\mu_i^2}$, Eq 10 converges to:

$$\mathcal{E}_i = sign\left(\frac{\mu_i}{\sigma_i}\right) = sign(\mu_i).$$ (42)

Substituting the function $\mathcal{E}_i$ in Eq 9, we can obtain Eq 17.  □

## F    Proof of Theorem 5

*Proof.* When $B \ll \frac{\pi \sigma_i^2}{2\mu_i^2}$, we have the approximate results in Eq 39 from Theorem 3. Substituting $\mathcal{E}_i$ and $\mathcal{B}_{noise}$ using Eq 39 and 14, and defining $\Delta L_{max}$ as in 19, the optimal loss improvement in Eq 8 can be expressed as:

$$\Delta L_{opt}(B) = \frac{\frac{1}{2} \cdot \frac{2B}{\pi} \sum_i \sum_j \frac{\mu_i^2 \mu_j^2}{\sigma_i \sigma_j}}{\sum_i H_{i,i} + \frac{2B}{\pi} \sum_i \sum_j \begin{cases} \frac{\mu_i \mu_j}{\sigma_i \sigma_j} & i \neq j \\ 0 & i = j \end{cases} H_{i,j}}$$

$$= \frac{\sum_i \sum_j \frac{\mu_i^2 \mu_j^2}{\sigma_i \sigma_j}}{\frac{\pi \sum_i H_{i,i}}{B} + 2 \sum_i \sum_j \begin{cases} \frac{\mu_i \mu_j}{\sigma_i \sigma_j} & i \neq j \\ 0 & i = j \end{cases} H_{i,j}} \tag{43}$$

$$= \frac{1}{\frac{\mathcal{B}_{noise}}{B} + 1} \frac{\sum_i \sum_j \frac{\mu_i^2 \mu_j^2}{\sigma_i \sigma_j}}{2 \sum_i \sum_j \begin{cases} \frac{\mu_i \mu_j}{\sigma_i \sigma_j} & i \neq j \\ 0 & i = j \end{cases} H_{i,j}}$$

$$= \frac{\Delta L_{max}}{\frac{\mathcal{B}_{noise}}{B} + 1} \leq \Delta L_{max}$$

Following the Appendix D in [24], this allows both the total number of steps and data examples processed to still be written as

$$S = \int \left(1 + \frac{\mathcal{B}_{noise}}{B}\right) ds$$
$$E = \int (\mathcal{B}_{noise} + B) ds \tag{44}$$

Therefore the conclusion in [24]

$$S_{min} = \int ds$$
$$E_{min} = \int \mathcal{B}_{noise} ds \tag{45}$$

and Eq 20, 21 still hold.

$\square$

## G    Variable Estimation for Data/Time Efficiency Relationship Equation

When considering $S$, $E$, $S_{min}$ and $E_{min} > 0$, Eq 20 can be simplified to

$$\left(\frac{S}{S_{min}} - 1\right)\left(\frac{E}{E_{min}} - 1\right) = 1$$
$$SE - S_{min}E - SE_{min} + \cancel{S_{min}E_{min}} = \cancel{S_{min}E_{min}}$$
$$S_{min}E + SE_{min} = SE$$
$$\frac{S_{min}}{S} + \frac{E_{min}}{E} = 1 \tag{46}$$
$$\frac{1}{S} = -\frac{E_{min}}{S_{min}}\frac{1}{E} + \frac{1}{S_{min}}$$

so a linear fit to the relationship between $\frac{1}{S}$ and $\frac{1}{E}$ can estimate $S_{min}$ and $E_{min}$.

## H    Additional Experiments Results

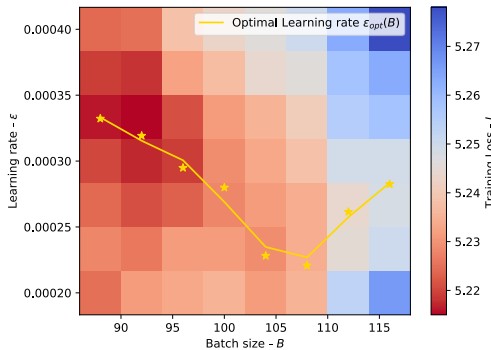

Figure 6: Finer-grained grid search results for the experiments shown in Figure 4(b).

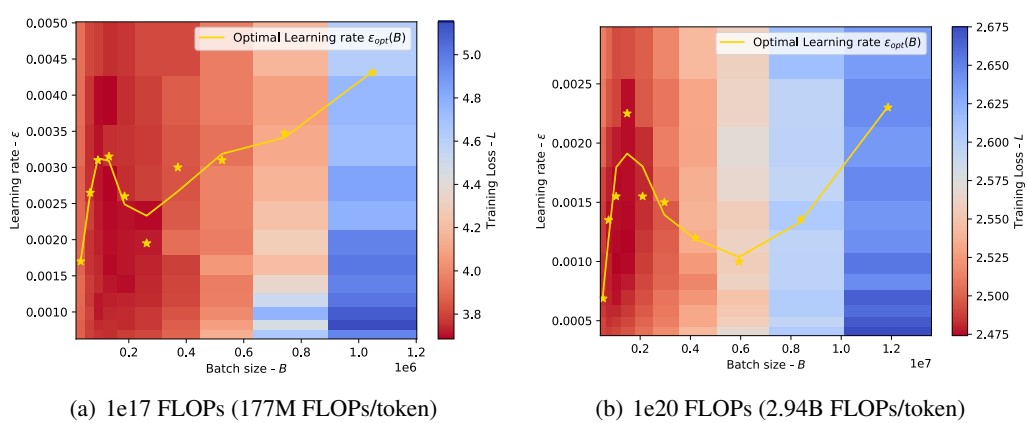

(a) 1e17 FLOPs (177M FLOPs/token)

(b) 1e20 FLOPs (2.94B FLOPs/token)

Figure 7: The optimal learning rates, based on the results presented in the Deekseek paper [63], align with our theorems.

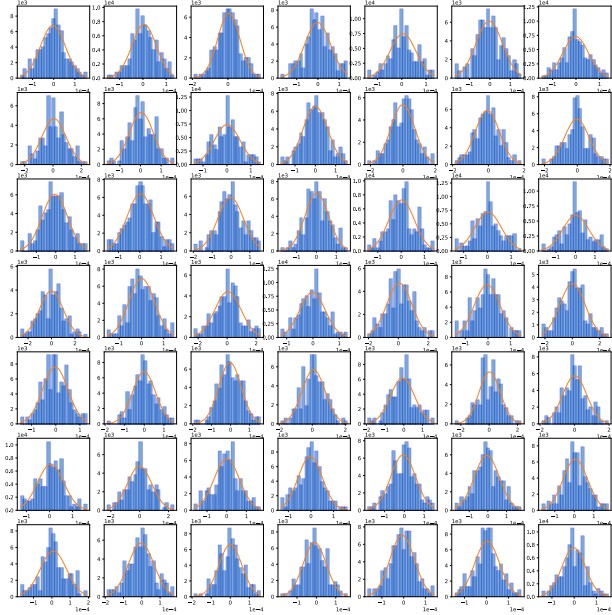

Figure 8: Examples of gradient distributions observed during the training of an MoE structure model, which approximate Gaussian distributions.

