# OpenReview forum: "Surge Phenomenon in Optimal Learning Rate and Batch Size Scaling"
_NeurIPS.cc/2024/Conference — NeurIPS 2024 poster_

### Official Review · Reviewer_PmLA · 2024-07-07

**Soundness:** 3
**Presentation:** 3
**Contribution:** 2
**Rating:** 5
**Confidence:** 3

**Summary:**

The paper analyzes the intricate relationship between optimal learning rate and batch size scaling for adaptive optimizers, such as Sign SGD and Adam. Building on prior analysis for SGD by McCandlish (2018), this work reveals a non-monotonic relationship between optimal learning rates and batch size. The optimal learning rate initially increases, reaches a peak, and then decreases, eventually saturating to a non-zero value, referred to as the surge phenomenon. These theoretical predictions are validated through experiments in both image classification and NLP tasks.

**Strengths:**

* The paper provides new insights into the relationship between optimal learning rate and batch size for Adam optimizers. The drop in the optimal learning rate after the peak is new, to the best of my knowledge, and is relevant given that Adam is the default optimizer choice.
* Prior results suggesting a square root scaling for Adam are reproduced, further supporting the findings.

**Weaknesses:**

* At times, the paper assumes that the reader is well-versed with the prior work of McCandlish (2018). For instance, lines 129-131. It would be helpful to reiterate prior results to motivate the analysis.
* The empirical results are not very convincing. If we only consider the experiments (for instance Figure 4), without any reference to theoretical results, the surge phenomenon does not seem appreciable. A finer learning rate search has to be performed to demonstrate the surge phenomenon clearly. In Figure 4(b), the optimal learning rate is oscillating around two points. It is unclear if this is due to the surge phenomenon or just random fluctuations. I would request the authors to help me understand their empirical results better.
* The theoretical results are derived for sign SGD, while it's known that Adam parameters beta1 and beta2 are crucial hyperparameters. It's unclear why the theoretical results can be generalized to Adam.

**Questions:**

* What is the practical implication of the decrease in the optimal learning rate after the peak?
* Given that the peak shifts through training, can the authors propose guidelines for scaling the learning rate and batch size?
* What is the intuition behind the training speed and data efficiency relationship being the same as the SGD case?
* Why the ResNet model (Figure 3) is trained with random 10k samples at every epoch? This should affect the overall results. Also, why this experiment is performed with sign SGD only?
* If the results of Figure 2(b) and 2(c) are combined, then it should predict (1) a drop in the optimal learning rate, (2) saturation, (3) increase, and finally saturation. How does this result align with the main result (Figure 1)?

**Limitations:**

* The theory is built on the quadratic approximation of the loss function. In the last few years, it has been established that modern neural networks are typically trained at large learning rates (Edge of Stability, see Refs. [1-2]), which cannot be captured using quadratic approximations of the loss [3].
* Gaussian distribution for gradients is assumed for the theoretical analysis, whereas it is known that the gradient distribution is not Gaussian, and this is precisely why Adam performs better than SGD in language modeling. It is unclear whether the results hold for such settings.


[1] Gradient descent on neural networks typically occurs at the edge of stability
Cohen et al. (2021)
arXiv:2103.00065

[2] Adaptive gradient methods at the edge of stability
Cohen et al. (2022)
arXiv:2207.14484

[3] Self-stabilization: The implicit bias of gradient descent at the edge of stability
Damian et al. (2022)
arXiv:2209.15594

[4] Linear attention is (maybe) all you need (to understand transformer optimization)
Ahn et al. (2022)
arXiv:2310.01082

---

> ### Author Rebuttal · Authors · 2024-08-06
>
> We are truly grateful for the time you have taken to review our paper and your insightful reviews. Below, we address your comments in detail.
>
>
> > W1.
>
> A1. Due to paper page limitation, we were unable to delve deeply into references [1-2]. We acknowledge this limitation and will provide a more comprehensive analysis in a future edition.
>
> Regarding the analysis of training speed and data efficiency in lines 129-131, we provide a detailed exposition in A6. In summary, Eq. 20 plays a pivotal role in both [1] and [2], particularly in its modification of the scaling law in [2]. If we demonstrate that Eq. 20 holds in the context of Adam, the modified scaling law conclusion in [2] naturally follows.
>
>
> > W2.
>
> A2. Following your suggestion, we conducted finer search and presented the results in Fig. 2 of the attached file.  The outcomes from these extended experiments provided clearer result which aligns with our theoretical analyses.
>
>
> > W3.
>
> A3. In Appendix A of our paper, we explain why the theoretical results in sign SGD can be generalized to Adam. In both our original experiments in our paper (section 3.1 and 3.3) and our supplementary Experiment-1, we set both hyperparameters beta1 and beta2 to non-zero values, and still observed the proposed surge phenomenon in the experimental results. More details and results be found in the global rebuttal and the appendix.
>
>
> > Q1.
>
> A4. Our motivation for investigating the scaling law of learning rates with respect to batch sizes is to gain deeper insights into the training dynamics of deep learning models. This understanding can aid in fine-tuning hyperparameters, enhancing convergence speed, and avoiding exhaustive grid searches. Leveraging prior knowledge that the optimal learning rate decreases after reaching its peak, researchers and engineers can effectively adjust the learning rate to achieve more efficient training.
>
>
> > Q2.
>
> A5. Given that the peak shifts during training, a straightforward approach is to periodically adjust hyperparameters to redetermine the scaling law, including hyperparameters $\epsilon_{max}$ and $\mathcal B_{noise}$. Our observation may serve as inspiration for developing adaptive learning rate or batch size methods in the future, which can effectively harness this knowledge.
>
>
> > Q3.
>
> A6. Section 2.3 of [1] demonstrates that integrating its Eq. 2.7 during the optimization process results in Eq. 2.11 (Eq. 20 in our paper). This implies that if the delta loss after each optimization step follows $\frac{\Delta L_{max}}{1 + \frac{\mathcal{B}}{B}}$, similar conclusions to Eq. 2.11 can be derived. Consequently, proving Eq. 18 ensures the validity of Eq. 20. This suggests that, Adam and SGD only affect the value of $\Delta L_{max}$, and do not involve other interference within the relation between the delta loss and batch size.
>
>
> > Q4.
>
> A7. Considering the substantial training costs associated with grid search in our experiments, we utilize random 10k samples to mitigate these expenses. This approach accelerates model convergence while yielding consistent conclusions. Notably, when attempting to use the original dataset, we encountered the need for larger models to fit the data, resulting in significantly longer training times—far beyond what we can afford.
>
> Beyond the considerations of training costs, selecting a distinct value for the hyperparameter $\beta_{i}$ does not compromise the validity of our conclusions. We provide a comprehensive discussion on the relationship between signSGD and Adam in Appendix-A. Furthermore, our paper and supplementary file include experiments that explore outcomes related to the Adam scenario. More details and results can be found in our paper (section 3.1 and 3.3) and our supplementary Experiment-1.
>
>
> > Q5.
>
> A8. Thank you for your observation. In Fig. 2, we deliberately chose scenarios with higher loss to emphasize the decrease and subsequent increase learning rates parts. In these cases, the batch size corresponding to the peak of the curve (the red dashed lines) is relatively small in this case, leading to the grid search not fully capturing the rising range of the curve. However, our other experiments, documented in both the main paper and the supplementary file, reveal curves that more are closely align with the trend depicted in Fig. 1. We include these scenarios to gain a clearer understanding of the underlying dynamics.
>
>
> > L1.
>
> A9. Thank you for your insightful suggestions. Our experiments, including the supplementary ones, demonstrate that the conclusions drawn from the second-order expansion approximation of the loss function effectively predict the surge phenomenon observed in most mainstream scenarios. In our final paper revision, we will thoroughly review and discuss the referenced articles. Additionally, we recognize the potential of exploring higher-order approximations with respect to scaling laws as promising future work.
>
>
> > L2.
>
> A10. As observed in Fig. 2(a) of our paper and Fig. 4 in our supplementary file, the distribution of gradients along the dataset direction approximately adheres to Gaussian distributions. Furthermore, as demonstrated in Equation 11, this equation indeed possesses a functional extremum as long as the distribution satisfies ($f(B) \propto B$). This implies that, under broader conditions, both the batch size and learning rate still exhibit a surge during the scaling process. We appreciate the suggestions provided in L1-2 and plan to explore this issue more thoroughly in future work, incorporating the ideas discussed therein.
>
>
> Thanks again for appreciating our work and providing constructive suggestions. We hope you will consider raising your score. Please let us know if you have further questions.
>
> ---
>
> [1] An Empirical Model of Large-Batch Training, 2018, McCandlish et al. (reference [25] in our paper)
>
> [2] Scaling Laws for Neural Language Models, 2020, Kaplan et al. (reference [26] in our paper)

---

> > ### Comment · Reviewer_PmLA · 2024-08-11
> >
> > I thank the authors for their extensive rebuttal.
> >
> > I have gone through the rebuttal plots and I still don't find the empirical results regarding the surge phenomena convincing. For instance, Figure 2 of the rebuttal does not look similar to the claimed curve from Figure 1. I would request the reviewers to further help me understand the difference.

---

> > > ### Author Response · Authors · 2024-08-12
> > >
> > > Thank you for your time and feedback. The surge phenomenon refers to the behavior where the optimal learning rate increases with an increase in batch size, then decreases, and finally increases slowly. This is illustrated by the orange solid line in Figure 1 of the original paper: rising from 0 to 0.4e8, falling from 0.4e8 to 1.5e8, and rising again from 1.5e8 to 4e8. Supplementary Material Figure 2 provides a finer grid search in the batch size range of 80-120 and learning rate range of 0.002-0.004, expanding Figure 4(b) from the original paper. This highlights that the observed phenomenon of decreasing and then increasing within the batch size range of 80-120 in Figure 4(b) is not due to randomness. From a broader perspective in Figure 4(b), as the batch size increases, the optimal learning rate first increases from 0 to 65.8, then decreases from 65.8 to 105, and finally slowly increases from 105 to 120, displaying a similar trend to Figure 1 in the original paper. Although the specific **batch size** ranges for each phase may vary with different models and datasets, the general trend described by the surge phenomenon is consistent across these scenarios.

---

> > > > ### Comment · Reviewer_PmLA · 2024-08-12
> > > >
> > > > I thank the authors for the clarification. The rebuttal result is clear to me now. I am currently keeping my score but I may increase it later.
> > > >
> > > > Again, I would like to thank the authors for their time and effort.

---

> > > > > ### Author Response · Authors · 2024-08-12
> > > > >
> > > > > Dear Reviewer PmLA,
> > > > >
> > > > > We sincerely appreciate your valuable comments and the time and effort you invested in reviewing our paper. We will ensure to incorporate these suggestions into the final manuscript.
> > > > > Thank you very much for considering raising the score! Should there be any further points that require clarification or improvement, please know that we are fully committed to addressing them promptly.

---

> > > > > ### Author Response · Authors · 2024-08-14
> > > > >
> > > > > Dear Reviewer PmLA,
> > > > >
> > > > > Thank you for increasing the score. If you have any other comments in the future, we would be delighted to provide further clarification and revisions as needed.

---

### Official Review · Reviewer_ASLs · 2024-07-12

**Soundness:** 3
**Presentation:** 1
**Contribution:** 2
**Rating:** 6
**Confidence:** 3

**Summary:**

The paper presents a heuristic analysis of the scaling of the optimal learning rate with batch size for Adam-style optimizers in the framework of [1]. The analysis is accompanied by experiments to support the predictions. Notably the authors demonstrate that the optimal learning rate can decrease with batch size in a certain range, a phenomenon that had not been identified previously and which is not present for SGD. The analysis also recovers the square-root scaling rule in the small batch size regime identified in other work.

[1] McCandlish, S., Kaplan, J., Amodei, D., & Team, O. D. (2018). An empirical model of large-batch training. arXiv preprint arXiv:1812.06162.

**Strengths:**

The paper tackles a practically important problem using a mix of heuristic theory and experiments. The prior literature on linear scaling rules with large batch training applies to SGD but not to Adam style optimizers which are the dominant optimizers for transformers. The surge phenomenon is interesting and novel.

**Weaknesses:**

The presentation is not great. A lot is assumed from [1], but it would make reading easier to make things more self-contained. Equations like Eq. 22 should be better explained and plots like in Fig 3 are hard to parse. The process for making the Fig. 3 plot is unclear. It is unclear what is going on in Figure 1 between the solid and dashed Adam curves. The takeaways and consequences for a practitioner are unclear. Minor: the Latex parentheses look sloppy.

[1] McCandlish, S., Kaplan, J., Amodei, D., & Team, O. D. (2018). An empirical model of large-batch training. arXiv preprint arXiv:1812.06162.

**Questions:**

My questions are related to the perceived weaknesses

1. In Fig. 1 how is the solid Adam curve generated? Shouldn't the curve eventually asymptote? It doesn't seem like that from the plot.
2. How is Fig. 3 generated? The linear fits look very strange.
3. Do the results suggest using any modification of the square root scaling rule in practice? If so it would be helpful to have a comparison to understand potential benefit.
4. Is there any characterization of the **intermediate** (i.e. large but not infinite batch) behavior of the scaling?

It would be helpful to have an analog of Figure 1 for the empirical data.

**Limitations:**

Yes.

---

> ### Author Rebuttal · Authors · 2024-08-06
>
> Thank you for taking the time to review our paper and for your valuable insights. We address your comments as follows.
>
> > W1. The presentation is not great. A lot is assumed from [1], but it would make reading easier to make things more self-contained.
>
> A1. Due to page constraints, we currently include only essential assumptions and conclusions from related work. In the final version of our paper, we aim to make the contents more self-contained and enhance our presentation.
>
>
> > W1. It is unclear what is going on in Figure 1 between the solid and dashed Adam curves.
> > Q1. In Fig. 1 how is the solid Adam curve generated? Shouldn't the curve eventually asymptote? It doesn't seem like that from the plot.
>
> A2. The solid line of Fig. 1 in our paper is a schematic diagram that we created based on Eq. 13 and 17, while the dashed line of Fig. 1 is an illustration for Eq. 13. According to Eq. 11 and 13, there exists an extreme value of learning rate, resulting in a tendency to increase first and then decrease. As $B$ continues to increase, $\mathcal E_{i}(B)$ grows slowly according to Eq. 10 and 17, and finally asymptotically approaching a value, causing the solid line to gradually rise after a downtrend. For better visualization, we only showed part of the figure without plotting the final asymptote. Since the specific values of the downward and upward inflection points are related to the current loss and are hard be determined, the schematic diagram is only presented here for illustration purposes. Furthermore, in supplementary Experiment-3, we can provide additional evidence to support our conclusions by visualizing the experiment results in accordance with the trend shown in Fig. 1. More details can be found in the global rebuttal. We will provide more detailed explanations in the final version of our paper.
>
>
> > W1. Equations like Eq. 22 should be better explained and plots like in Fig 3 are hard to parse. The process for making the Fig. 3 plot is unclear.
> > Q2. How is Fig. 3 generated? The linear fits look very strange.
>
> A3. Eq. 22 is derived from Eq. 20-21, and the detailed derivation is in Appendix G. We will add more explanations in the final version of our paper. In Fig. 3, the left part is the grid search results of batch sizes and learning rates, while the right part is the fitted curve of Eq. 22 using the S and E data from corresponding (batch size, best learning rate) pairs in the left part. We will improve the presentation of the experiments.
>
>
> > W1. The takeaways and consequences for a practitioner are unclear.
> > Q3. Do the results suggest using any modification of the square root scaling rule in practice? If so it would be helpful to have a comparison to understand potential benefit.
>
> A4. In our experiments (see Fig. 2-4 in our paper and also the figures in the attached file), our fitted curve outperforms the square root scaling (corresponding to the alpha=0.5 curve). We have discussed the benefits of this approach in Section 3.3 and global rebuttal. Notably, while square root scaling may perform well in small batch size scenarios, it is essential to recognize that in large batch size scenarios, the optimal learning rate may no longer scale linearly and could decrease. For practitioners, we recommend following the papers [1-2] to approximate $\mathcal B_{noise}=B_{crit}$ using scaling law $B_{crit} = \frac{B^*}{L^{\frac{1}{\alpha}}}$, then use one simple search for a pair of (batch size, optimal learning rate) to determine the last hyper-parameter $\epsilon_{max}$. Subsequently, the conclusion of our paper can be applied to avoid costly grid-search. We will clarify the process in the final version of our paper.
>
>
> > Q4. Is there any characterization of the intermediate (i.e. large but not infinite batch) behavior of the scaling?
>
> A5. In our supplementary Experiment-3, we validated our theory by referring to publicly available experiment results from another paper, which included data on scaling in the intermediate transition phase. We have also visualized these results to further support our findings. More details and results be found in the global rebuttal and the appendix file.
>
>
> > W1. Minor: the Latex parentheses look sloppy.
>
> A6. We will improve the readability of our paper.
>
>
> Thanks again for appreciating our work and for your constructive suggestions. We hope you will consider raising your score. Please let us know if you have further questions.
>
> ---
>
> [1] An Empirical Model of Large-Batch Training, 2018, McCandlish et al. (reference [25] in our paper)
>
> [2] Scaling Laws for Neural Language Models, 2020, Kaplan et al. (reference [26] in our paper)

---

> > ### Comment · Reviewer_ASLs · 2024-08-13
> >
> > Thank you for the responses and additional results. I will increase my score.

---

> > > ### Author Response · Authors · 2024-08-14
> > >
> > > Dear Reviewer ASLs,
> > >
> > > We would like to express our gratitude for your valuable feedback and for increasing the score of our paper. Your perceptive remarks and recommendations have significantly enhanced the quality of our work, and we sincerely value the time and effort you dedicated to reviewing our submission.

---

### Official Review · Reviewer_JT4e · 2024-07-12

**Soundness:** 3
**Presentation:** 2
**Contribution:** 2
**Rating:** 6
**Confidence:** 3

**Summary:**

The paper gives an optimal choice of learning rate and batch size for neural networks.  Different from the previous results on SGD-style optimizers. The authors give such solutions for Adam-style-optimizers.

**Strengths:**

1. Batch size and learning rate will affect the performance largely and cost a lot to select a good one. It is important to understand the optimal batch size and learning rate for Adam-style optimizers.

2. The experimental results match the theorem proposed by the authors.

**Weaknesses:**

1. It seems that Lemma 1 can only apply to quadratic problems. In the appendix, the relation is approximately equal. But in Lemma 1, it becomes "equal" without any further assumptions.

2.  It is unclear how to select the optimal batch size or learning rate based on the theorem because either S, E or $\mu,\sigma$ is hard to estimate for a large network.

**Questions:**

1. Why does the theorem apply to general network optimization?

2. How to make the result of the theorem in use?

---

> ### Author Rebuttal · Authors · 2024-08-06
>
> We are grateful for the time and effort you have taken in reviewing our paper and for your thoughtful feedback. Here, we respond to your comments.
>
>
> > W1. It seems that Lemma 1 can only apply to quadratic problems. In the appendix, the relation is approximately equal. But in Lemma 1, it becomes "equal" without any further assumptions.
> > Q1. Why does the theorem apply to general network optimization?
>
> A1. We would like to clarify that Lemma 1 can be applied to **ALL** gradient descent-based machine learning processes. In this context, the quadratic term arises from the Taylor expansion of the loss with updated parameters, **NOT** from quadratic problems. General network optimizations all have the form of gradient descent-based learning. Here we use the "equal" sign for simplicity. The similar analysis and simplification were also proposed in OpenAI's paper[1] Eq. 2.4-2.7. We will improve the clarification and the symbol, and add the reference in Lemma 1 in the final version of our paper.
>
>
> > W2. It is unclear how to select the optimal batch size or learning rate based on the theorem because either S, E or &mu;, &sigma; is hard to estimate for a large network.
> > Q2. How to make the result of the theorem in use?
>
> A2. Our work extends the conclusions on SGD optimizers in OpenAI's papers[1-2], focusing on Adam-style optimizers. Though in experiments we used grid-search to validate our conclusions, in practice the scaling law only contains two unknown hyper-parameters $\epsilon_{max}$ and $\mathcal B_{noise}$. Following papers [1-2], $\mathcal B_{noise}=B_{crit}$ can be efficiently approximated using scaling law $B_{crit} = \frac{B^*}{L^{\frac{1}{\alpha}}}$. We only need one simple search for a pair of (batch size, optimal learning rate) to determine the last hyper-parameter $\epsilon_{max}$. Therefore, the costly grid-search can be avoided. We will improve the presentation of our paper and add this clarification.
>
>
> Thanks again for appreciating our work and for your constructive suggestions. We hope you will consider raising your score. Please let us know if you have further questions.
>
> ---
>
> [1] An Empirical Model of Large-Batch Training, 2018, McCandlish et al. (reference [25] in our paper)
>
> [2] Scaling Laws for Neural Language Models, 2020, Kaplan et al. (reference [26] in our paper)

---

> > ### Comment · Reviewer_JT4e · 2024-08-12
> >
> > Can you provide some experimental results showing that A2 is possible because the current results are based on grid-search on all hyperparameters?

---

> > > ### Author Response · Authors · 2024-08-12
> > >
> > > We appreciate your questions. To demonstrate the feasibility of A2, we follow the procedure outlined in A2 to determine the optimal learning rates relative to batch sizes, using the MoE experiment as an example (the data used below are from Fig. 1 in the supplementary file).
> > >
> > > ---
> > >
> > > **Step 1 Approximate $\epsilon_{opt}$ using a small amount of data**
> > >
> > > To demonstrate the generalizability of A2, here we try 3 starting points to derive the scaling laws separately. For each of the 3 different batch sizes (3145728, 4718592, 6291456), we search 5 (learning rate, training loss) pairs to fit $\epsilon_{opt}$, in which, the training loss uses $normalized(L) = \frac{L - L_{min}}{L_{max} - L_{min}}$ .
> > >
> > > |                |          BS-1         |   BS-2  |   BS-3  |
> > > |:--------------:|:---------------------:|:-------:|:-------:|
> > > |    Token BS    |        3145728        | 4718592 | 6291456 |
> > > |    LR (1e-4)   | $normalized(L)$ |         |         |
> > > |      1.0       |        1.0000         | 1.0000  | 1.0000  |
> > > |      2.0       |        0.4837         | 0.5134  | 0.5558  |
> > > |      3.0       |        0.1786         | 0.2144  | 0.2269  |
> > > |      4.0       |        0.0000         | 0.0000  | 0.0269  |
> > > |      5.0       |        0.0828         | 0.0186  | 0.0000  |
> > > |        A       |        0.0946         | 0.0782  | 0.0688  |
> > > |       -B       |        0.7995         | 0.7168  | 0.6658  |
> > > | $\epsilon_{opt}=-\frac{B}{2A}$ |      **4.2257**       | **4.5831**  | **4.8387**  |
> > >
> > >
> > > ---
> > >
> > > **Step 2 Approximate $\epsilon_{max}$ using a pair of (B, $\epsilon_{opt}$)**
> > >
> > > Following the referenced papers [2] and using their scaling law $B_{crit}=\frac{B^*}{L^{1/\alpha}}$, we can get $B_{crit} \approx 10^7$. We then use one pair of (B, $\epsilon_{opt}$) to approximate $\epsilon_{max}$, and the results are shown below. The last column is the actual result from grid search for comparison.
> > >
> > > |               |   BS-1  |   BS-2  |   BS-3  | AdEx 1 |
> > > |:-------------:|:-------:|:-------:|:-------:|:------:|
> > > |    Token BS   | 3145728 | 4718592 | 6291456 |        |
> > > | $\epsilon_{opt}$ (1e-4) |  4.2257 |  4.5831 |  4.8387 |        |
> > > | $\epsilon_{max}$ (1e-4) |  **4.9374** |  **4.9001** |  **4.9628** | 4.9363 |
> > >
> > > ---
> > >
> > > **Step 3 Use the fitted values to predict the best learning rates $\epsilon_{opt}$ for different batch sizes**
> > >
> > > Using ($B_{crit}$, $\epsilon_{max}$) from Step 2, we can predict the optimal learning rate $\epsilon_{opt}$ for different batch sizes with the formula $\epsilon_{opt}=\frac{\epsilon_{max}}{\frac{1}{2}(\sqrt{\frac{\mathcal B_{noise}}{B}} + \sqrt{\frac{B}{\mathcal B_{noise}}})}$ in our paper. The predicted $\epsilon_{opt}$ values and the actual values from grid search are shown below. We can see that the relationsip between the optimal learning rate and batch size derived from these 3 batch sizes all align with the actual results from the grid search.
> > >
> > > | Token BS | $\epsilon_{opt}$ (1e-4) |         |         |         |
> > > |:--------:|:------------:|:-------:|:-------:|:-------:|
> > > |          |     BS-1     |   BS-2  |   BS-3  | AdEx 1  |
> > > |  196608  |    1.3654    | 1.3551  | 1.3724  | 1.3651  |
> > > |  294912  |    1.6561    | 1.6436  | 1.6646  | 1.6558  |
> > > |  786432  |    2.5799    | 2.5604  | 2.5931  | 2.5793  |
> > > |  1572864 |    3.3982    | 3.3724  | 3.4156  | 3.3974  |
> > > |  3145728 |    4.2257    | 4.1937  | 4.2474  | 4.2247  |
> > > |  4718592 |    4.6180    | 4.5831  | 4.6417  | 4.6170  |
> > > |  6291456 |    4.8140    | 4.7776  | 4.8387  | 4.8129  |
> > > |  9437184 |    4.9361    | 4.8988  | 4.9614  | 4.9350  |
> > > | 12582912 |    4.9018    | 4.8647  | 4.9269  | 4.9007  |
> > >
> > >
> > > Therefore, by starting with one batch size and its corresponding five learning rate results, we can approximate the scaling law between optimal learning rates and batch sizes.

---

> > > > ### Comment · Reviewer_JT4e · 2024-08-13
> > > >
> > > > Thank you for the additional experiments. I believe adding the additional experiments helps me understand better. I have increased my score.

---

> ### Author Response · Authors · 2024-08-13
>
> Dear Reviewer JT4e,
>
> We would like to express our sincere gratitude for your valuable feedback and for increasing the score of our paper. Your insightful comments and suggestions have greatly improved the quality of our work. We will ensure to incorporate these suggestions into the final manuscript, and we truly appreciate the time and effort you have put into reviewing our submission.

---

### Official Review · Reviewer_ytaJ · 2024-07-13

**Soundness:** 3
**Presentation:** 3
**Contribution:** 3
**Rating:** 7
**Confidence:** 3

**Summary:**

This work provides a scaling law between learning rate and batch size for Adam. Namely, this work finds that the optimal learning rate increases and then decreases as the batch size becomes larger; and, the peak of this curve corresponds to the trade-off between training speed and data efficiency.

**Strengths:**

- The paper is well motivated. While prior works studying the relationship between batch size and learning rate have focused on SGD, this work focuses on Adam (which is more popular / widely used).
- The paper is written clearly and is well organized. I appreciate the “summary” notes included by the authors
- The paper includes empirical evidence for CV and NLP tasks to support theoretical claims.

**Weaknesses:**

- I think this paper could benefit from experiments with more popular architectures for the NLP tasks (e.g. maybe it would be useful to include some experiments on tasks with llama or mistral models).
- I also think it would be useful to have experiments with more datasets. Recent work shows that the data itself matters. E.g., for fine tuning LLMs, many factors wrt the data (e.g., data quality, variable sequence lengths, deduplicated data) can affect training. It would be interesting to see if the surge phenomena is agnostic to these factors or not.

**Questions:**

For NLP related tasks, the number of tokens in a batch might vary. Would the surge phenomena still apply for learning rate vs. num tokens?

**Limitations:**

It would be interesting to see results on a larger variety of modern and widely used datasets and architectures.

---

> ### Author Rebuttal · Authors · 2024-08-06
>
> Thank you for your thorough review and constructive suggestions. We address your comments in the following paragraphs.
>
> > W1. I think this paper could benefit from experiments with more popular architectures for the NLP tasks (e.g. maybe it would be useful to include some experiments on tasks with llama or mistral models).
>
> A1. We provided additional experimental analyses on both sparse MoE and dense Transformer models for NLP tasks. The details can be found in the global rebuttal. For the sparse MoE structure, we experimented on the fine-grained MoE model with shared experts[1], which is a popular model and has a similar structure to the well-known Mistral-MoE[2]. As for the dense model, we validated our proposed conclusions by referring to another paper's experiments[3]. Please refer to the global rebuttal for the configurations, results, and analyses of these experiments.
>
>
> > W2. I also think it would be useful to have experiments with more datasets. Recent work shows that the data itself matters. E.g., for fine tuning LLMs, many factors wrt the data (e.g., data quality, variable sequence lengths, deduplicated data) can affect training. It would be interesting to see if the surge phenomena is agnostic to these factors or not.
>
> A2. In the global rebuttal and our attached files, we incorporated additional datasets to enhance the robustness of our findings. Specifically, the Experiment-1 utilized the RedPajama-v2 dataset, and Experiment-3 used a combination of Deepseek's in-house data and OpenWebText2. Please refer to global rebuttal for datasets' details.  We acknowledge that data-related factors may affect training. However, due to their complexity and abundance, we leave the impact of these factors on surge phenomena as future work, which could lead to a separate, valuable research endeavor.
>
>
> > Q1. For NLP related tasks, the number of tokens in a batch might vary. Would the surge phenomena still apply for learning rate vs. num tokens?
>
> A3. In our paper and previous analysis[4], "batch size" is actually "token batch size", which considers the total number of tokens in a batch.  In our experiments, we use the packing strategy to splice the data in each batch, so the number of tokens in all batches can be considered equal. We will address the misuse of this terminology in the final version.
>
>
> Thanks again for appreciating our work and for your constructive suggestions. Please let us know if you have further questions.
>
> ---
>
> [1] DeepSeekMoE: Towards Ultimate Expert Specialization in Mixture-of-Experts Language Models, 2024, Dai et al.
>
> [2] Mixtral of Experts, 2024, Jiang et al.
>
> [3] DeepSeek LLM Scaling Open-Source Language Models with Longtermism, 2024, Bi et al.
>
> [4] An Empirical Model of Large-Batch Training, 2018, McCandlish et al. (reference [25] in our paper)

---

> > ### Comment · Reviewer_ytaJ · 2024-08-12
> >
> > Thank you for your response. I acknowledge that I have read the author's response. I will keep my score.

---

> > > ### Author Response · Authors · 2024-08-13
> > >
> > > Dear Reviewer ytaJ,
> > >
> > > We sincerely appreciate your valuable comments and the time and effort you invested in reviewing our paper. We will ensure to incorporate these suggestions into the final manuscript. We believe that these changes have significantly improved our paper.
> > > Once again, we express our sincere gratitude for your valuable contribution to our work.

---

### Author Rebuttal · Authors · 2024-08-06

Dear reviewers,

Thank you for your reviews and constructive suggestions. We have incorporated additional experimental analyses to strengthen our conclusions.


**Experimental Analyses**
We made experimental analyses on both sparse MoE and dense structures. For the sparse MoE structure, in Experiment-1, we experimented on a fine-grained MoE model with shared experts[1], which has a similar structure to Mistral-MoE[2], a representative of the sparse models. As for the dense model, in Experiment-2 and Experiment-3, we validated our proposed conclusions with a more fine-grained experiment on DistilGPT-2, as well as an experiment from another paper[3], which showed aligned results with our theorems. The following is our specific experimental configurations and results.

---

**Experiment - 1**

We performed a grid search on a 500M parameter MoE model with 16 experts using the RedPajama-v2 dataset.

| Key                 | Value          | Key                 | Value          |
|---------------------|----------------|---------------------|----------------|
| `VOCAB_SIZE`        | 32000          | `Adam Beta1`        | 0.9            |
| `N_LAYERS`          | 12             | `Adam Beta2`        | 0.999          |
| `MAX_SEQ_LENGTH`    | 4096           | `Weight decay`      | 0.00001        |
| `D_MODEL`           | 768            | `MAX_GRAD_NORM`     | 1              |
| `N_HEADS`           | 12             | `AUX_LOSS_ALPHA`    | 0.1            |
| `HEAD_DIM`          | 64             | `WSD warmup`        | 2000           |
| `EXPERTS_TOTAL_DIM` | 12288          | `WSD decay ratio`   | 0.1            |
| `TOP_K_ROUTER`      | 4              | `Dataset`           | RedPajama-v2   |
| `N_EXPERTS`         | 16             |


In the experiments, we aimed to find the largest best learning rate across token batch sizes (token bs, i.e., considering the number of tokens in a batch). For simplicity, we fitted the final training loss at different learning rates using a quadratic function of the form $f(x) = Ax^2+Bx+C$, which served as a good approximation. Subsequently, we utilized the extremum $-\frac{B}{2A}$ of the quadratic function to estimate the optimal learning rate.

We have presented the fitted curve in the attached file Fig. 1. Our method aligns well with the observed data points. Other methods, including linear scaling and square root scaling, cannot align with the best learning rates.

The results reaffirm our original conclusions: the optimal learning rate first increases and then decreases as batch size grows. This further validates our scaling law between learning rate and batch size for Adam.

---

**Experiment - 2**

We conducted a detailed grid search on DistilGPT-2 using the ELI5 category dataset. The results are plotted as Fig. 2 in the attached file.

---

**Experiment - 3**

Additionally, we note that a publicly available paper[3] provides grid search results that align with our theoretical predictions. We have plotted the fitted curves in Fig. 3 of the attached file. For better visualization, we removed the data points with large loss values in the experimental results. Notably, the $B_{crit}$ value is approximately 0.1M for (a) and 1.5M for (b), estimated using the Scaling Law $B_{crit} = \frac{B^*}{L^{\frac{1}{\alpha}}}$ , matching the peak position of the optimal learning rate observed in their experiments. This observed trend also conforms to the pattern depicted by the orange solid line in Fig. 1 of our paper, further validating our theoretical results.

---

[1] DeepSeekMoE: Towards Ultimate Expert Specialization in Mixture-of-Experts Language Models, 2024, Dai et al.

[2] Mixtral of Experts, 2024, Jiang et al.

[3] DeepSeek LLM Scaling Open-Source Language Models with Longtermism, 2024, Bi et al.

---

### Decision · Program_Chairs · 2024-09-25

**Decision:**

Accept (poster)

**Comment:**

This paper elucidate the connection between optimal learning rates and batch sizes for Adam-style optimizers through both theoretical analysis and extensive experiments (I quoted). All reviewers agree to accept this paper because (1) the method is well motivated, (2) experimental results are impressive and support the theory, (3) providing new insights to the scaling rule in Adam, etc. I agree with them, and thus recommend an accept.